# A Ruthenium(II) *N*-Heterocyclic Carbene (NHC) Complex with Naphthalimide Ligand Triggers Apoptosis in Colorectal Cancer Cells via Activating the ROS-p38 MAPK Pathway

**DOI:** 10.3390/ijms19123964

**Published:** 2018-12-09

**Authors:** Yasamin Dabiri, Alice Schmid, Jannick Theobald, Biljana Blagojevic, Wojciech Streciwilk, Ingo Ott, Stefan Wölfl, Xinlai Cheng

**Affiliations:** 1Institute of Pharmacy and Molecular Biotechnology, Heidelberg University, Im Neuenheimer Feld 364, 69120 Heidelberg, Germany; dabiri@stud.uni-heidelberg.de (Y.D.); alice.schmid@stud.uni-heidelberg.de (A.S.); jannick.theobald@gmail.com (J.T.); blagojevic@stud.uni-heidelberg.de (B.B.); wolfl@uni-hd.de (S.W.); 2Institute of Medicinal and Pharmaceutical Chemistry, Technische Universität Braunschweig, Beethovenstraße 55, 38106 Braunschweig, Germany; w.streciwilk@tu-bs.de (W.S.); ingo.ott@tu-bs.de (I.O.)

**Keywords:** naphthalimide-metal complex conjugates, *N*-heterocyclic carbene, mitochondria, ROS, p38 MAPK, apoptosis, cancer

## Abstract

The p38 MAPK pathway is known to influence the anti-tumor effects of several chemotherapeutics, including that of organometallic drugs. Previous studies have demonstrated the important role of p38 both as a regulator and a sensor of cellular reactive oxygen species (ROS) levels. Investigating the anti-cancer properties of novel 1,8-naphthalimide derivatives containing Rh(I) and Ru(II) *N*-heterocyclic carbene (NHC) ligands, we observed a profound induction of ROS by the complexes, which is most likely generated from mitochondria (mtROS). Further analyses revealed a rapid and consistent activation of p38 signaling by the naphthalimide-NHC conjugates, with the Ru(II) analogue—termed MC6—showing the strongest effect. In view of this, genetic as well as pharmacological inhibition of p38α, attenuated the anti-proliferative and pro-apoptotic effects of MC6 in HCT116 colon cancer cells, highlighting the involvement of this signaling molecule in the compound’s toxicity. Furthermore, the influence of MC6 on p38 signaling appeared to be dependent on ROS levels as treatment with general- and mitochondria-targeted anti-oxidants abrogated p38 activation in response to MC6 as well as the molecule’s cytotoxic- and apoptogenic response in HCT116 cells. Altogether, our results provide new insight into the molecular mechanisms of naphthalimide-metal NHC analogues via the ROS-induced activation of p38 MAPK, which may have therapeutic interest for the treatment of various cancer types.

## 1. Introduction

Increasing evidence has proven 1,8-naphthalimides as promising candidates for the treatment of cancer, with several such derivatives (e.g., amonafide and mitonafide) being tested in clinical trials against various solid and soft tumors [1]. Despite their potent anti-cancer activity, the clinical application of most of these compounds is hampered due to the toxic side effects [1,2]. Accordingly, several strategies have been developed to modify the naphthalimide ring in order to improve the anti-tumor effects and lower its toxicity [2]. This has led to the synthesis of various naphthalimide-based conjugates, such as metal complexes with naphthalimide ligands [2]. In this regard, naphthalimide-gold(I) phosphine complexes, whose synthesis was inspired by pervious observations on the lead compound auranofin, have shown an increase in the overall cellular uptake and in the nuclear accumulation of gold(I) as compared with the naphthalimide-free analogues [3]. Other interesting examples are the naphthalimide-based ruthenium(II) arene complexes, showing enhanced cancer cell selectivity, which is possibly achieved by the simultaneous action of naphthalimide as a DNA intercalator along with the ability of ruthenium(II) in binding proteins [4]. More recently, 1,8-naphthalimides containing a metal *N*-heterocyclic carbene (NHC) moiety have been synthesized [5,6]. These conjugates are shown to act via both interaction with DNA—related to the naphthalimide structure—as well as metal-based mechanisms, such as the inhibition of the thioredoxin reductase (TrxR) [5,6]. 

Among the molecular mechanisms that are involved in the anti-cancer efficacy of organometallic drugs is the mitogen activated protein kinase (MAPK) pathway. MAPKs encompass three signaling cascades; (i) extracellular signal-related kinases (ERKs), (ii) the c-Jun N-terminal kinase (JNK), and (iii) p38 MAPK, all of which have key roles in cellular proliferation and survival [7]. p38 MAPK has been repeatedly implicated in cancer therapy and its activation is shown to be necessary for cancer cell death triggered by a number of chemotherapeutic agents [8]. We and others have shown a determinant role for the activation of p38 signaling in the pro-apoptotic effects of metal-based drugs, such as cisplatin [9,10,11], auranofin [12] as well as gold-containing NHC complexes [13]. However, reports regarding the influence of naphthalimide derivatives on MAPKs are sparse. As an example, a novel amonafide analogue has been shown to down-regulate ERK1/2 and p38 via TAK1 inhibition, leading to its anti-inflammatory effects [14]. 

In continuation to the aforementioned studies, a series of 4-ethylthio-1,8-naphthalimide conjugates has been recently synthesized, carrying rhodium(I)- and ruthenium(II) NHC fragments as metal units [15]. The compounds were found to interact with DNA via an intercalation mechanism and they were able to trigger strong anti-proliferative effects in MCF-7 breast cancer and HT-29 colon carcinoma cells [15]. In this article, we describe more detailed analyses on the molecular mechanisms underlying the anti-tumor effects of Rh(I)- and Ru(II) naphthalimide-NHC compounds, as well as the metal-free ligand, designated as MC7, MC6, and MC5, respectively. All of the complexes showed potent anti-proliferative effects against various breast- and colorectal cancer (Colorectal cancer (CRC)) cell lines, but exhibited mild toxicity in human foreskin fibroblasts (HFFs). Using HCT116 CRC cells, we have assessed the involvement of reactive oxygen species (ROS) and p38 MAPK signaling in the mode of action of these molecules. We observed elevated intracellular- and mitochondrial ROS production, and a remarkable activation of p38 MAPK in response to naphthalimide-NHC analogues, with no clear regulation of other members of the MAPK family. Additionally, the modulation of ROS and p38α by anti-oxidants and either chemical inhibitors or siRNA, respectively, led to a significant reduction in the pro-apoptotic and growth inhibitory functions of the Ru(II) derivative. Our findings propose p38 signaling as a novel anti-cancer target of organometallic complexes with naphthalimide ligands.

## 2. Results

### 2.1. MC5, MC6, and MC7 Exhibit Cytotoxic Effects in Tumor Cells of Different Tissue Origins

To determine the cytotoxicity of the naphthalimide-NHC complexes (Figure 1A) in different cancer models, the cell lines of breast-(MCF-7 and MDA-MB-231) and colorectal (HCT116) tissue origins were treated with increasing concentrations of the compounds for three incubation periods (24, 48, and 72 h), after which Sulforhodamine B (SRB) assay was performed. The metal-free naphthalimide ligand—MC5—as well as the rapid apoptosis inducer, raptinal [16] were included as references. In all of the investigated cell lines, cellular survival was found to decrease time- and concentration-dependently in response to the three compounds, with HCT116 cells showing the highest sensitivity to metal-containing analogues in the first 24 h (Figure 1B). After 48 and 72 h of incubation with MC6 and MC7, cell viabilities became comparable among MCF-7 and HCT116 cells, while they remained substantially higher in the case of MDA-MB-231 at most of the tested concentrations (Figure 1B). Moreover, MC5 was found to be particularly active against MCF-7 breast cancer cells, but not in HCT116 and MDA-MB-231. This is in good agreement with the previous report, showing significantly lower IC_50_ values of metal-free naphthalimide species in MCF-7 as compared to that of HT-29 CRC cells [15]. Notably, the compounds exhibited extremely low cytotoxic effects in HFF cells, even at the highest tested concentrations, which might suggest the preferential toxicity of the naphthalimide-NHC conjugates towards cancer cells (Appendix A).

We also tested the cytotoxicity of the three analogues in the context of p53-deficiency or mutant p53 using the p53-null HCT116 and HT-29 cell lines, respectively. As shown in Appendix A, mutant p53 harboring HT-29 cells are less sensitive to the cytotoxic effects of the compounds at most tested concentrations and time points. 

### 2.2. MC5, MC6, and MC7 Inhibit Cell Cycle Progression in HCT116 CRC Cells

To shed light on the mechanism that is responsible for the inhibitory activity of the compounds on cellular viability, we sought to assess changes in the cell cycle regulation. HCT116 cell line was selected for further investigation based on its higher susceptibility to the MC6- and MC7-mediated cytotoxic effects (Figure 1B). A 24 h post-treatment analysis of the DNA content revealed a G1 arrest in response to treatments (Figure 2A). Although all three compounds caused a significant increase in the G1 phase cell population as compared to mock (0.1% DMSO), this effect was found to be more pronounced in the case of the Rh(I) analogue (MC7), followed by the metal-free ligand (MC5), and finally the Ru(II) complex (MC6) (Figure 2B). 

Additionally, we examined the mRNA levels and protein expression of p21, an inhibitor of the complexes of cyclin D and cyclin-dependent kinases (CDKs), which have a key role in the G1 to S phase transition [17]. As expected, p21 mRNA and protein levels were induced after 24 h of treatment, with no statistically significant difference being found across the three compounds (Figure 2C–E).

### 2.3. Intracellular- and Mitochondrial ROS Levels of HCT116 Cells Are Differentially Induced by the Three Naphthalimide-NHC Complexes

The involvement of excessive ROS generation has been repeatedly mentioned in the anti-cancer mechanisms of organometallic drugs, including that of metal NHC complexes [18]. We therefore sought to evaluate the influence of the three compounds on intracellular ROS formation in HCT116 cells using dihydroethidium (DHE) staining. After 24 h of treatment with various concentrations of each compound, we found that all of the complexes produced a modest but consistent increase in ROS levels, which occurred concentration-dependently (Figure 3A,B). When comparing the three molecules, the highest fold change (1.7) was observed after treatment with 50 μM of the metal-free ligand (MC5), followed by Ru(II) (MC6) and Rh(I) (MC7) compounds which caused a 1.2-fold increase in ROS levels at the highest used concentrations, 12 and 50 μM, respectively (Figure 3A). Pre-treatment with the ROS scavengers, *N*-acetyl-l-cysteine (NAC) and reduced glutathione (GSH) clearly prevented the MC6-triggered ROS production (Figure 3C). To gain further insight into the source of ROS, we sought to analyze the role of mitochondrial respiration and included co-treatment with either the complex I inhibitor, rotenone, or the uncoupling agent, CCCP. As shown in Figure 3C, blocking complex I was found to significantly decrease MC6-induced ROS, whereas co-treatment with CCCP (2.5 μM, 2 h) caused a slight increase in the total cellular ROS levels. 

One of the main sources of intracellular ROS is mitochondria, known as mitochondrial ROS (mtROS), which are produced in the form of superoxide anions (O_2_^−^) as a by-product of oxidative metabolism [19]. To obtain further insight into the ROS inducing ability of naphthalimide-NHC conjugates, we performed a live cell analysis of MitoSox Red staining. In contrast to intracellular ROS levels, mtROS production was found to have the highest induction with the Ru(II) complex (MC6) at all of the tested time points, followed by MC5, and finally the Rh(I) analogue (Figure 4A,B and Appendix A). As early as 3 h, mtROS were elevated up to 3.1-fold upon treatment with 12 µM of MC6 and reached the maximum after 12 h (5.4-fold), followed by a slight decrease at 24 h (Figure 4B). A similar decrease could be also observed with a concentration of 50 µM of MC5 (Figure 4B). Such a reduction after long-term treatments or higher concentrations might be due to the activation of anti-oxidant defense mechanisms by cancer cells, as previously reported for gold(I) NHC complexes [13].

To further elucidate the source of ROS, Mito TEMPO, which is a mitochondria-specific ROS scavenger, was pre-incubated with the cells 2 h before MC6 treatment for 24 h. As shown in Figure 4C, anti-oxidant treatment was capable of reducing mitochondrial superoxide levels induced by 12 µM of MC6.

The rich photophysical properties of naphthalimides make them useful tools for monitoring their uptake and localization in living cells [2]. Using fluorescence microscopy, we detected a clear mitochondrial accumulation of all the three compounds in HCT116 cells after 4 h of treatment (Figure 4D). This may explain the stronger effect of the compounds on mtROS generation rather than that of cytosolic ROS.

### 2.4. Naphthalimide-NHC Derivatives Impact Mitochondrial Membrane Potential (MMP) in Different Ways

MtROS production is determined by a number of factors, one of which is MMP (Δψm) [20]. Flowcytometric analysis of Δψm in HCT116 cells revealed that, among the three complexes, MC5 and MC6 have the lowest and the highest potentials at all the tested time points, respectively (Appendix A and Figure 5A). It has been proposed by a “redox-optimized ROS balance hypothesis” that physiological ROS signaling occurs at optimized MMP levels, whereas oxidative stress can happen at either extreme (low or high) of Δψm [21]. This is in line with our results, showing the highest mtROS production in case of MC6 and MC5, which exhibit the most oxidized and reduced redox potentials, respectively. Accordingly, MC7, whose MMP does not move far from the basal levels (Appendix A and Figure 5A), shows only a moderate increase in mtROS generation, as compared to those of the other two compounds (Appendix A and Figure 5A).

Bcl-xL is known to govern the integrity of the mitochondrial outer membrane through protecting it from Bax-induced permeabilization, which leads to the release of cytochrome *c* and activation of caspases [22]. In this regard, we observed a concentration-dependent decrease in the protein expression of the pro-survival Bcl-2 member, Bcl-xL after 24 h of treatment, with MC6 and MC7 showing the highest and lowest reduction, respectively (Figure 5C,D). This was in parallel to the transcriptional activation of the apoptogenic factors, *Bax* and *Bad* (Figure 5B). The pro-apoptotic function of Bad is known to be mediated via its interaction with Bcl-2/Bcl-xL, which neutralizes the pro-survival activity of the latter proteins, thereby sensitizing cells to apoptosis [23]. In support of this, we observed elevated Bad protein levels upon 24 h of treatment with all three compounds at the indicated concentrations (Figure 5C,D). Of note, it has been repeatedly shown that only the unphosphorylated Bad is able to heterodimerize with Bcl2/Bcl-xL and that phosphorylation of the protein at either of the three serine residues, S112, S136, and S155 sequesters Bad away from mitochondrial membrane [23]. We therefore evaluated Bad phosphorylation status in response to the compounds and observed a reduction in two of the aforementioned phosphorylation sites (S112 and S136), suggesting the potential role of Bad in promoting cell death in response to the three analogues (Figure 5C,D). Taken together, all the above findings demonstrate the involvement of ROS and mitochondrial death pathway in the anti-tumor effects of the naphthalimide-NHC conjugates.

### 2.5. p38 MAPK Signaling Is Activated in Response to Naphthalimide-NHC Complexes, with the Ru(II) Analogue Exhibiting the Strongest Effect

Several reports have demonstrated the activation of p38 signaling by a number of organometallic drugs, including gold(I) [13]- and Rh(I) [24] NHC complexes. MtROS generation, on the other hand, has been repeatedly associated with MAPK activation, leading to inflammatory responses, apoptosis, and autophagy [20]. We thus evaluated the involvement of this signaling molecule in the mode of action of naphthalimide-NHC analogues in HCT116 cells. After 24 h of treatments, the levels of phospho-p38 MAPK (pp38 MAPK; T180/Y182) were clearly increased across the three compounds, with the Ru(II) analogue (MC6) showing the highest fold change (~7) at a concentration of 12 μM (Figure 6A,B). The activation of p38 signaling by the Rh(I) and Ru(II) complexes was further confirmed as its downstream genes, *ATF2* and *Stat1* were found to be transcriptionally activated after 24 h of treatment with the respective concentrations of the compounds (Figure 6C). A time-dependent analysis of pp38 MAPK (T180/Y182) protein levels upon treatment with 12 μM of MC6 showed that p38 activation occurs as early as 1 h and it persists over the test period of 24 h (Figure 6D,E). Additionally, we observed a rapid and consistent activation of ATF2 in response to MC6, as determined by increased levels of its phosphorylated form (pATF2; T71) (Figure 6D,E).

Next, we aimed to assess whether p38 activation is reproducible in cancer cell lines other than HCT116 CRC cells. As shown in Figure 6F,G, MC6 and MC7 were able to induce the levels of phospho-p38 MAPK (pp38 MAPK(T180/Y182)) in the breast cancer cell lines, MCF-7 and MDA-MB-231, however with a much lower efficiency in the case of the latter. With regards to this, it has been reported that breast cancer cell lines with a triple-negative (ER negative, PR negative and HER-2 negative) molecular profile exhibit enhanced expression and activity of p38 MAPK, which has been correlated with poor prognosis and survival in patients [25]. Thus, the minor effect on p38 in MDA-MB-231 is most likely due to the endogenous higher activity. In this context, p38 may act as a tumorigenic factor rather than a tumor suppressor. Therefore, its inhibition rather than its activation is shown to exert anti-proliferative effects in invasive breast cancer models [26].

p38 and other members of the MAPK family of proteins, ERK and JNK, have been shown to crosstalk at several levels, determining whether the cell survives or undergoes apoptosis in response to chemotherapeutic drugs [27,28]. We therefore evaluated the phosphorylation status of ERK and JNK using immunoblotting, as well as protein ELISA-microarray analysis [29]. We observed a mild inhibitory effect of the compounds, in particular MC6 and MC7, on the phospho-activation of ERK (Appendix A), however the difference was not found to be statistically significant. With regards to JNK, 24 h treatment with increasing concentrations of the compounds did not show a clear alteration in the phosphorylated protein levels (Appendix A), which most probably rules out the involvement of JNK signaling in the pro-apoptotic response of HCT116 cells to naphthalimide-NHC analogues. 

### 2.6. Anti-Proliferative and Pro-Apoptotic Effects of the Ru(II) Naphthalimide-NHC Complex in HCT116 Cells Proceed via p38 MAPK Signaling, Involving ROS

To further elucidate the role of p38 in the anti-tumor properties of MC6, which triggered the strongest p38 activation across the three complexes (Figure 6A,B), we knocked-down the expression of *p38α* (*MAPK14*) in HCT116 cells while using siRNA (Figure 7D). As determined by SRB cytotoxicity assay, upon 24 h treatment with 12 μM of MC6, cellular viability was significantly increased in cells that were transfected with anti-*p38α* siRNA as compared to that of the negative control (siRNA NC) and non-transfected HCT116 cells (Figure 7A). We then aimed at investigating the influence of p38 knock-down on the MC6-mediated apoptosis using annexinV/propidium iodide (AV/PI) staining. As depicted in Figure 7B,C, 24 h treatment with 12 μM of MC6 resulted in the transition of cells through the early apoptotic (AV^+^/PI^−^) quadrant in HCT116 and HCT116 siRNA NC, as determined by ~1.8- and ~1.4-fold increase, respectively. This effect was found to be attenuated in response to the p38 knock-down, as the fold change of early apoptotic- as well as late apoptotic/necrotic (AV^+^/PI^+^) cells were significantly less than that of HCT116 and cells transfected with the non-targeting siRNA (Figure 7B). Similar results were obtained in the presence of two well-known chemical inhibitors of p38α, VX-702, and Ralimetinib. As shown in Figure 7E,F, the number of apoptotic cells upon 24 h of MC6 treatment was significantly decreased when combined with p38α inhibitors, as detected by AV/PI staining. Additionally, the pro-apoptotic response of HCT116 cells to MC6 was evaluated in the absence/presence of p38α activity using flow cytometric analysis of caspase 3 activation where significantly less cleaved caspase 3 levels were observed upon both knock-down of p38α (Figure 7G), as well as its pharmacological inhibition (Figure 7H,I). Moreover, we monitored the MC6-induced apoptosis using TUNEL assay and found significantly more apoptotic cells with 24 h of MC6 treatment, an effect that was clearly abrogated upon VX-702-mediated p38α inhibition (Figure 7J,K). 

Activation of the pro-apoptotic p38 MAPK signaling can happen as result of ROS generation [19], and it has been previously implicated in the anti-cancer mechanism of metal NHC complexes [13]. We hence sought to investigate whether pp38 MAPK induction and subsequent growth inhibitory/pro-apoptotic effects of MC6 are due the elevated ROS levels. As shown in Figure 8A,B, pre-incubation with the ROS scavengers, reduced GSH as well as NAC, blocked the MC6-induced p38 activation as well as apoptosis, determined by the absence of PARP and caspase 3 cleavages, markers of apoptotic cell death. With respect to MC6-mediated cytotoxicity, we observed increased cellular survival percentages with GSH treatment, however, the difference failed to reach statistical significance (Figure 8C). Similarly, pre-treatment with a mitochondria-targeted scavenger, Mito TEMPO, hampered the effects of MC6 on p38 activation and PARP cleavage (Figure 8D,E), as well as its cytotoxicity (Figure 8F) in HCT116 cells. We further confirmed the protective effect of anti-oxidant treatment on MC6-mediated apoptosis by quantification of cleaved caspase 3 levels using flow cytometry. As shown in Figure 8G,H, pre-treatment of HCT116 cells with GSH and NAC rescued the MC6-mediated caspase 3 cleavage. In view of these findings, we propose that the in vitro anti-tumor activity of MC6 in HCT116 CRC cells may be regulated through mtROS-induced activation of the p38 MAPK pathway.

## 3. Discussion

Regulation of the redox state plays an important role in tumor cell survival. Although elevated ROS levels allow cancer cells to promote pro-tumorigenic signaling, excessive ROS production is usually associated with anti-tumorigenic pathways, which can trigger oxidative stress-induced cancer cell death [19]. The latter effects of ROS are mainly mediated through the ASK1/JNK and ASK1/p38 axes, leading to cell cycle arrest, growth inhibition, and apoptosis [19]. Here, we demonstrate that a novel Ru(II) naphthalimide-NHC complex is able of causing a remarkable increase in mtROS generation, which in turn activates the pro-apoptotic p38 signaling in HCT116 CRC cells. 

The p38 pathway is a major regulator of stress responses, influencing various biological processes, including cellular proliferation and survival [8]. Studies have demonstrated that the role of p38 MAPK signaling in cancer therapy is contextual, depending on the nature of the stimuli, cancer type, as well as the status of other MAPKs (ERKs and JNKs) [30,31]. On the one hand, p38 MAPK activation mediates the sensitivity of tumor cells to a variety of chemotherapeutics, for example, cisplatin [9,10,11,32], oxaliplatin [33], and auranofin [12]. In particular, cisplatin has been reported to induce apoptosis in HCT116 CRC cells through the ROS-p38α axis downstream of p53 activation [10]. On the other hand, cancer cell lines with a high basal expression of pp38 MAPK tend to lose the tumor-suppressing functions of this molecule, possibly because of the inability to further activate p38 MAPK in response to anti-cancer treatments [26]. In light of this, we detected a clear induction of pp38 MAPK (T180/Y182) levels in response to all three naphthalimide-NHC derivatives in HCT116 and MCF-7 cells, with the Ru(II) analogue (MC6) exhibiting the strongest effect. However, MC6 was hardly able to promote p38 activation in the triple-negative MDA-MB-231 breast cancer cells in which the basal levels of phospho-p38 are elevated (Figure 6F,G). This is in line with a reduced cytotoxic response of MDA-MB-231 as compared with that observed for HCT116 and MCF-7 (Figure 1B), suggesting that the lack of p38 activation renders this cell line resistant to the anti-proliferative effects of naphthalimide-NHC conjugates. In this regard, several reports have shown the beneficial outcome of p38 MAPK inhibition rather than its activation for the treatment of invasive breast cancer models. For example, it has been proposed that p38 MAPK inhibition has synergistic effects with cisplatin for the treatment of breast cancer through the activation of ROS-mediated JNK signaling [27]. Our results suggest an anti-tumorigenic role for p38 signaling in response to MC6 as its genetic as well as chemical inhibition attenuates the cytotoxic- and pro-apoptotic effects of the compound in HCT116 cells (Figure 7). However, this requires further investigation in other cellular contexts, in particular, in cancer cells with enhanced basal p38 MAPK phosphorylation. To further address whether other MAPKs, including JNK and/or ERK signaling, are implicated in these processes, we also analyzed the regulatory state of these kinases in response to treatment with the three naphthalimide-NHC derivatives, as, for instance, JNK activation and/or ERK inhibition, which has been previously reported in the anti-cancer efficacy of several other organometallic drugs [27,34,35]. We, however, were not able to observe a profound regulation of MAPKs other than p38α in response to the naphthalimide-NHC analogues (Appendix A), further demonstrating the important role of p38α activation in sensitizing tumor cells to MC6-mediated apoptosis. 

In this study, we show an essential role for mtROS generation in the cytotoxicity of naphthalimide-NHC analogues. There are two main sources of the signaling-associated ROS in cells: (i) membrane bound-NADPH oxidases (known as NOX enzymes) and (ii) mitochondrial electron transport chain (ETC) [19]. Our results demonstrate that intracellular ROS levels are minimally affected by the naphthalimide-NHC derivatives, whereas mitochondrial superoxide production is markedly increased in response to the molecules, with the Ru(II) complex showing the highest fold change. MtROS have been frequently mentioned in the regulation of pro-inflammatory/apoptotic responses via multiple mechanisms, among others, is the activation of MAPKs [20]. In view of this, we observed that treatment with a mitochondria-targeted ROS scavenger and MC6 blocks the latter’s effect on p38 MAPK activity (Figure 8D,E), suggesting that mtROS generation may be acting upstream of MC6-mediated p38 induction. It is known that mitochondrial superoxide species sustain MAPK activity through the oxidation and inactivation of MAPK phosphatases (MKPs) [36]. In line with this, we found reduced MKP6 expression in response to the naphthalimide-NHC conjugates (data not shown), further supporting the involvement of mtROS in the induction of p38 signaling. Importantly, the cytotoxic and pro-apoptotic effects of MC6 were hampered upon general- and mitochondria-specific anti-oxidant treatments (Figure 8), implicating a ROS-mediated pathway underlying the in vitro anti-tumor efficacy of the complex. 

One suggested mode of action of metal NHC complexes and naphthalimide derivatives is via mitochondrial accumulation, and perturbations in the MMP (Δψm) [13,37,38]. We here report that Δψm is differentially regulated by the three naphthalimide-NHC conjugates, showing all possible options; a timely decrease in case of MC5, a transient increase in case of MC6 followed by a decrease at 24 h, and no significant change in case of MC7. With regards to the relationship between mitochondria-driven ROS and Δψm, the general concept is that more polarized membrane (high Δψm) is associated with greater ROS production [20]. This is consistent with the observation of MC6 showing the highest MMP along with the highest mtROS generation as compared to the other two complexes. However, it fails to explain the influence of MC5 treatment on mitochondrial parameters, wherein, despite reduced Δψm, mtROS production increases. These disparate observations reconcile by a “redox-optimized ROS balance” model proposed by Aon et al. who demonstrated that oxidative stress can occur at either extreme of MMP (high Δψm or low Δψm), meaning that under oxidizing conditions mtROS can increase because of the compromised anti-oxidant defense mechanisms [21]. Taking all the mentioned observations into consideration, we suggest that the mode of anti-cancer activity of naphthalimide-NHC compounds is most likely through mitochondrial localization, leading to ROS-mediated activation of the pro-apoptotic p38 MAPK pathway.

## 4. Materials and Methods

### 4.1. Chemicals and Antibodies

1-(3’-(4”-ethylthio-1”,8”-naphthalimid-*N*”-yl))-propyl-3-benzylimdazolium bromide (MC5), Dichloro[1-(3’-(4”-ethylthio-1”,”-naphthalimid-*N*”-yl))-propyl-3-benzyl-imidazol-2-ylidene] (η^6^-*p*-cymene)ruthenium(II) (MC6), and Chlorido[(η^2^, η^2^-cycloocta-1,5-diene)-1-(3’-(4”-ethylthio-1”,8”-naphthalimid-*N*”-yl))-propyl-3-benzyl imidazol-2-ylidene] rhodium(I) (MC7) were synthesized and purified, as previously described [6,15]. Raptinal, antimycin, GSH, NAC, Mito TEMPO, rotenone, CCCP, and JC-1 were purchased from Sigma-Aldrich (Steinheim, Germany). Pharmacological inhibitors of p38α, VX-702 and Ralimetinib (LY2228820) were from selleckchem (Munich, Germany) and DHE was from Biomol GmbH (Hamburg, Germany). MitoTracker Green, MitoSox Red, and the transfection reagent, Lipofectamine 3000 were obtained from Thermo Fischer (Darmstadt, Germany). Primary antibodies against Bcl-xL (#2764), pATF2 (T71; #9221), pp38 MAPK (T180, Y182; #9211), PARP (#9542), Bax (#5023), phospho-Bad sampler kit (#9105), caspase 3 (#9662), cleaved caspase 3 (#9661), pERK (T202, Y204; #5683), pStat1 (Y701; #7649), pStat1 (S727; #8826), pStat3 (Y705; #9145), pStat3 (S727; #9134), Stat1 (#9176), Stat3 (#9139), as well as anti-mouse and rabbit IgG horseradish peroxidase (HRP)-linked antibodies were purchased form Cell Signaling Technologies. Anti-vinculin (SC-73614), anti-p38 (SC-7972), and anti-ERK (SC-135900) were from Santa Cruz Biotechnology. 

### 4.2. Cell Culture

Human breast- (MCF-7, MDA-MB-231) and colon (HCT116 WT, HCT116 p53-/-, HT-29) cancer cell lines, as well as HFF were maintained in Dulbecco’s modified Eagle medium (DMEM)-GlutaMax (Gibco, Darmstadt, Germany) supplemented with 10% (*v*/*v*) FCS (Gibco) and 1% penicillin/streptomycin (*v*/*v*) (Gibco) and they were incubated under 5% CO_2_ and at 37 °C. All of the treatments were performed in the same media at the indicated conditions.

### 4.3. Cell Viability Assay

The SRB assay was employed to measure the inhibitory effects of MC5, MC6, and MC7 on the proliferation of HCT116 WT, HCT116 p53-/-, HT-29, MCF-7, and MDA-MB-231, as well as the influence of anti-oxidants (GSH and Mito TEMPO) and p38 knock-down on the toxicity of MC6-treated HCT116 CRC cells. Cells were seeded in either 96-well plates or 24-well plates at a density of 10,000- and 60,000 cells/well, respectively. At the end of treatments, plates were fixed with 10% ice-cold trichloroacetic acid (TCA), followed by incubation at 4 °C for at least one hour. Afterwards, well contents were washed three times with water and were dried at 60 °C. Next, 0.054% SRB sodium salt was added to the wells and incubated at room temperature for 30 mins. Plates were then washed three times with 1% acetic acid and were dried at 60 °C. Finally, the SRB dye was dissolved in 10 mM Tris (pH 10.5) and measurement was performed using the Tecan Ultra plate reader (Tecan, Crailsheim, Germany) at 535 nm absorbance wavelength.

### 4.4. Cell Cycle Analysis

For analysis of cell cycle phase distribution, 10^6^ cells/well were harvested, washed with PBS, then fixed with 70% ice-cold ethanol, and incubated at −20 °C overnight. After twice washing with ice-cold PBS (Gibco), cells were treated with 200 µg/mL of RNase (Sigma-Aldrich) for 30 mins at 37 °C, followed by 15 min incubation with PI (50 µg/mL) (Sigma-Aldrich) in the dark for nucleic acid staining. Samples were analyzed by FACSCalibur (Becton Dickinson, Franklin Lakes, NJ, USA) and data analysis was performed using the software, CellQuest™ Pro (Becton Dickinson). 

### 4.5. Intracellular ROS Measurement Using Flow Cytometry (FACS) and Live Cell Imaging

DHE and MitoSox Red were used for the detection of whole cell- and mitochondrial superoxide generation, respectively. For DHE staining, the HCT116 cells were seeded in 12-well plates at a density of 200,000 cells/well. After the indicated treatments, cells were incubated with phenol red-free DMEM containing 15 μM of DHE for 20 mins, harvested, and resuspended in fresh media. For MitoSox Red staining, 60,000 cells/well were seeded in 24-well plates. At the time points indicated, media were replaced with phenol red-free DMEM containing 5 μM of MitoSox Red for 15 mins. Cells were then harvested and resuspended in fresh media. FACS analysis was immediately performed using Guava easyCyte HT sampling flow cytometer (Guava Technologies, Hayward, CA, USA) and data were analyzed using GuavaSoft 3.1.1 software. 

Live cell imaging of mitochondrial ROS production was performed using the Incucyte ZOOM live cell analysis system (Essen BioScience, Broadwater Road Welwyn Garden City, United Kingdom). HCT116 cells were seeded in a 24-well plate at a density of 60,000 cells/well. On the following day, the cells were first incubated with 2 nM of MitoTracker Green dissolved in FCS- and pheno red-free DMEM for 20 mins, after which with fresh media containing 5 μM of MitoSox Red as well as the treatments. Time-lapse images were then taken every 30 mins for a period of 24 h. Data were analyzed using the Incucyte software 2016b. Briefly, the signal intensity was calculated based on a fluorescent area mask; with a top hat filter being used for excluding dead cells due to the higher auto-fluorescence. Nine pictures/well were used to determine the overall signal/well and each condition was performed in quadruplicates. 

### 4.6. Fluorescence Microscopy

For assessing mitochondrial localization of naphthalimide-NHC analogues, HCT116 cells were cultured in 12-well plates at a density of 150,000 cells/well. After the indicated treatments, the cells were washed with PBS and were incubated with 2 nM of MitoTracker Green dissolved in FCS- and phenol red-free media for 20 mins in order to stain mitochondria. Cells were then washed with PBS and were imaged using the BIOREVO fluorescence microscope (BZ9000, KEYENCE; Neu-Isenburg, Germany). 

### 4.7. Mitochondrial Membrane Potential Assessment

200,000 cells/well were seeded in 12-well plates. Treated cells were harvested and then incubated with phenol red-free DMEM containing 2 μM of the JC-1 dye for 15 mins in the dark. After resuspension in fresh media, FACS analysis was performed by Guava easyCyte HT sampling flow cytometer using GuavaSoft 3.1.1 software for data analysis. 

### 4.8. RNA Isolation, Reverse Transcription, and Quantitative Real Time (qRT) PCR

At the end of treatments, total RNA was isolated using QIAzol lysis reagent (Qiagen, Hilden, Germany) and the quality of RNA samples was determined by NanoDrop 2000 UV-Vis Spectrophotometer (Thermo Scientific, Darmstadt, Germany). cDNA synthesis was performed from 500–1000 ng of total RNA using ProtoScript II first strand cDNA synthesis kit (New England Biolabs, Frankfurt am Main, Germany), according to the manufacturer’s instructions. The thermal cycler q-Tower (Analytik Jena AG, Jena, Germany) was used to analyze gene expression levels. The amplification reaction solutions (5 μL) were prepared with 2.5× of ready to use master mix LightCycler^®^ 480 SYBR Green I (Roche, Mannheim, Germany), 1× of nuclease-free H_2_O and cDNA templates, as well as 1× of the following primer mixtures (Eurofins Genomics, Ebersberg, Germany): *CDKN1A* (5s: GACACCACTGGAGGGTGACT; 3as: CAGGTCCACATGGTCTTCCT), *Bax* (5s: GGGGACGAACTGGACAGTAA; 3as: CAGTTGAAGTTGCCGTCAGA), *Bad* (5s: GGTTCTGAGGGGAGACTGAGG; 3as: GCTTCCTCTCCCACCGTAGC, *ATF2* (5 s: CAGCGTTTTACCAACGAGGA; 3as: GA ATCTTGTTGGTGTTGGGGTC), *Stat1* (5 s: GGAAAAGCAAGCGTAA TCTTCAGG; 3as: GAATATTCCCCGACTGAGCC), and *vinculin* (as reference gene) (5s-CAGTCAGACCCTTACTCAGTG-3′; 3as-CAGCCTCATCGAAGGTAAGGA).

### 4.9. Immunoblotting

After the respective treatments, cells were lysed using a urea-based lysis buffer supplemented with multiple phosphatase/protease inhibitors, namely sodium orthovanadate, aprotinin, PMSF, pepstatin, and sodium pyrophosphate. Protein concentration was determined with Bradford reagent (Sigma-Aldrich). 20–50 µg of total protein was separated on 10% SDS-PAGE, then transferred onto a PVDF membrane (GE Healthcare, Munich, Germany), after which it was blocked with 5% (*w*/*v*) milk in TBST (Tris-Buffered Saline Tween-20) for at least 1 h. Membranes were then incubated with primary antibody solutions (diluted following the manufacturer’s instructions) overnight at 4 °C and were visualized by further incubation with the corresponding HRP-linked secondary antibodies (diluted at 1:5000 in 5% (*w*/*v*) milk TBST) for 1 h at room temperature, followed by three washing steps with TBST. Target proteins were finally detected by Western Lightning™ Plus ECl (Perkin Elmer, Waltham, USA) using the Fujifilm LAS-3000 imaging system and were quantified with ImageJ software.

### 4.10. Protein ELISA-Microarray Analysis

We have previously reported a detailed protocol on the assay [29]. Briefly, treated cells were lysed in a similar manner to that of immunoblotting. Proteins were then diluted 1:6 in a PBS-based buffer (containing 1 mM EDTA, 0.5% (*v*/*v*) Triton X-100, and 5 mM NaF). Protein concentration was assessed using the Pierce BCA Protein Assay kit (Thermo Fischer). The levels of phosphorylated ERK1/2 and JNK were quantified using sandwich ELISA microarrays that were based on the ArrayStrip platform (Alere Technologies GmbH, Jena, Germany). Signals were detected by the Arraymate reader (Alere Technology GmbH). Data were analyzed using the KOMA software [39] and were normalized to the amount of total proteins used.

### 4.11. siRNA-Mediated Knock Down

siRNA oligos against *p38α* (*MAPK14*) as well as the non-targeting siRNA were obtained from Thermo Fischer. HCT116 cells were transfected with 40 pmol of the respective siRNA diluted in 100 μL/well Opti-MEM Reduced Serum Medium (Gibco) and complexed with 1.5 μL/well of Lipofectamine 3000 in a 24-well plate and they were incubated for 20 mins at room temperature. Cell suspension was then added at a density of 60,000 cells/well and treatments were performed on the following day, as indicated. 

### 4.12. Cell Death Analysis (AV/PI Staining)

AV- and PI staining were performed in parallel for the detection of early apoptotic and late apoptotic/necrotic cells, respectively. 250,000 cells/well of the non-transfected HCT116, HCT116 cells transfected with either a negative control- or anti-*p38α* siRNA, as well as HCT116 cells in the absence/presence of p38α pharmacological inhibitors were harvested, washed with AV binding buffer (BD Biosciences, Heidelberg, Germany), and then resuspended in 50 μL binding buffer containing 5 μL of both FITC-conjugated AV and PI (BD Biosciences). Samples were incubated for 15 mins in the dark, after which they were resuspended in binding buffer up to 500 μL for FACS analysis, which was done using Guava easyCyte HT sampling flow cytometer. Data were analyzed by GuavaSoft 3.1.1 software. 

### 4.13. Caspase 3 Activation Assay

At the end of the treatments, 60,000 cells/wells were harvested, fixed, and permeabilized with 4% paraformaldehyde and 90% ice-cold methanol, respectively, as previously described [40]. Cells were then incubated overnight with cleaved caspase 3 antibody at a dilution of 1:800 and they were further incubated with secondary antibody (goat anti-rabbit Alexafluor 488-conjugated; Dianova, Hamburg, Germany) for 1 h at room temperature, thereafter analyzed using Guava easyCyte HT sampling flow cytometer and the GuavaSoft 3.1.1 software. 

### 4.14. TUNEL Assay

TUNEL assay was performed for the detection of DNA fragmentation as a characteristic of late stage apoptosis using Cell Meter TUNEL Apoptosis Assay Kit (Biomol GmbH, Hamburg, Germany). Briefly, 60,000 cells/well were harvested, fixed, and permeabilized by 4% paraformaldehyde and 0.2% Triton X-100, respectively. After washing with PBS, cells were incubated with the tunnelyte reaction mixture for 1 h at 37 °C, thereafter the fluorescence intensity was measured by the Tecan Ultra plate reader (Tecan) at an excitation/emission wavelength of 550/650 nm, respectively. Additionally, the cells were analyzed using the BIOREVO Fluorescence microscope (BZ9000, KEYENCE). 

### 4.15. Statistical Analyses

Data were analyzed using GraphPad Prism and Microsoft Excel. Densitometric analyses were performed by the ImageJ software. Statistical significance between the respective treatment and DMSO was determined by student’s *t*-test. Multiple comparisons were performed using either one-way or two-way ANOVA, followed by a post-hoc Tukey test as indicated. *p*-values less than or equal to 0.05, 0.01, 0.001, and 0.0001 are presented as *, **, ***, and *** on the figures, respectively.

## 5. Conclusions

Several studies have supported the therapeutic use of high ROS levels present in tumor cell lines for the induction of pro-apoptotic pathways. In this study, we show that altered cellular redox balance induced by a novel Ru(II) naphthalimide-NHC complex—MC6—plays an important role in its in vitro anti-tumor efficacy, most likely via mediating the activation of p38 MAPK signaling. Nonetheless, several issues remain to be addressed with regards to the activation of ROS-p38 axis by naphthalimide-NHC analogues, such as: (i) if the pro-apoptotic functions of MC6-induced p38 MAPK could be extrapolated to other cellular contexts, in particular, the ones with enhanced p38 expression and (ii) whether the MC6-triggered elevation in mtROS levels is through an increase in ROS production at the mitochondrial respiratory chain complexes and/or an alteration in cellular/mitochondrial anti-oxidant systems, for instance, superoxide dismutases (SODs), glutathione peroxidases (GPXs), and thioredoxins (TRXs). 

## Figures and Tables

**Figure 1 ijms-19-03964-f001:**
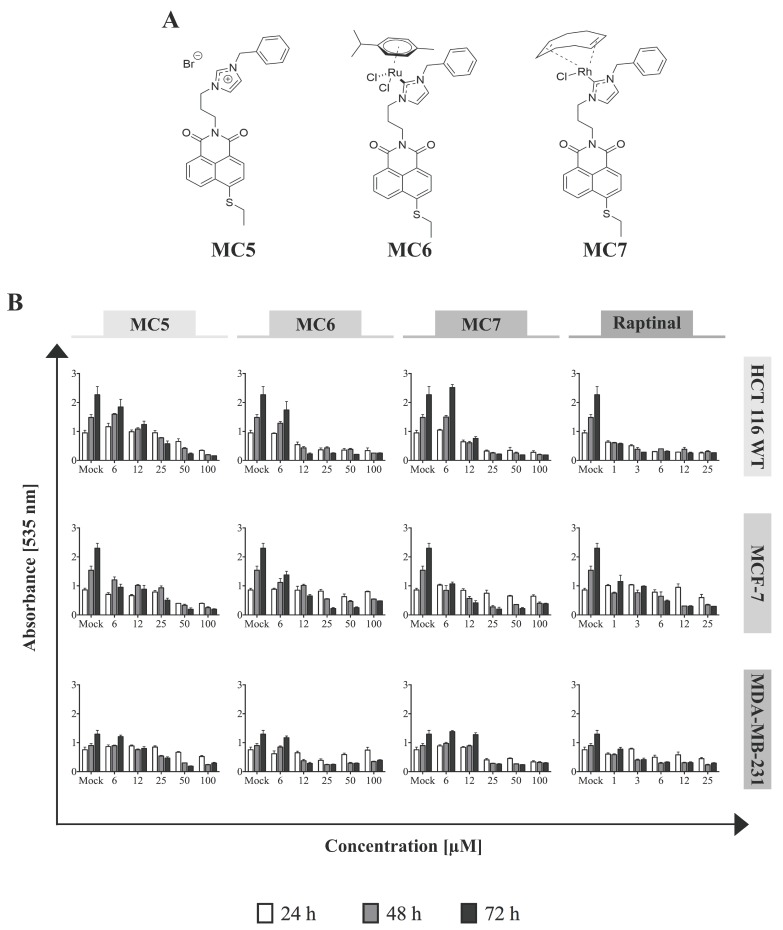
Naphthalimide-*N*-heterocyclic carbene (NHC) analogues exhibit cytotoxic effects against human breast- and colon cancer cells. (**A**) Chemical structures of the compounds; (**B**) Increasing concentrations of each of the complexes, as well as the rapid apoptosis inducer, raptinal [16] (as positive control) were applied to the different cell lines and Sulforhodamine B (SRB) assay was performed after 24, 48, and 72 h of treatment. The Ru(II)- and Rh(I)-containing complexes show the highest and the least efficacy against HCT116 and MDA-MB-231, respectively, in most tested concentrations. 0.1% DMSO-treated cells served as mock. Data represent mean ± SD of three independent experiments, each was done in quadruplicates.

**Figure 2 ijms-19-03964-f002:**
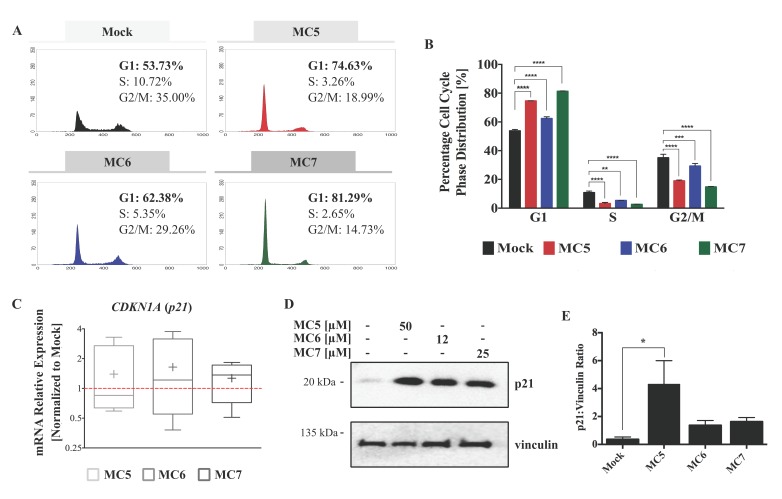
Naphthalimide-NHC analogues induce cell cycle arrest and p21 expression in HCT116 CRC cells. (**A**) Representative histogram plots show the distribution of cell cycle phases in HCT116 cells treated with either 0.1% DMSO (as mock) or the three complexes, MC5, MC6, and MC7 at a concentration of 50, 12, and 25 μM, respectively for 24 h; (**B**) All of the analogues were found to induce a G1 phase arrest as compared to mock treatment. Comparison of the percentage cell population of G1, S, and G2/M phases between mock and each of the three complexes was performed by two-tailed student’s *t*-test. Error bars represent the SD of two biological replicates, one of which is depicted in (**A**); (**C**) p21 mRNA levels are up-regulated in response to 24 h of treatments, analyzed by qRT-PCR. Relative expression was calculated using the ∆∆ Ct method where the Ct values of p21 were normalized to those of the housekeeping gene (vinculin). Lower and upper ends of the bars indicate the minimum and maximum values, respectively, and the “+” in the middle represents the mean. Error bars ± SD; *n* = 4; (**D**) p21 protein levels upon 24 h of treatment with the three complexes at the indicated concentrations, determined by immunoblotting; (**E**) Densitometric quantification of p21 bands normalized to those of the loading control (vinculin). Error bars indicate the SEM of two biological replicates, one of which is presented in (**D**). Multiple comparisons were made using one-way ANOVA test and a post-hoc Tukey test. *, **, ***, and **** denote *p*-values less than or equal to 0.05, 0.01, 0.001, and 0.0001, respectively.

**Figure 3 ijms-19-03964-f003:**
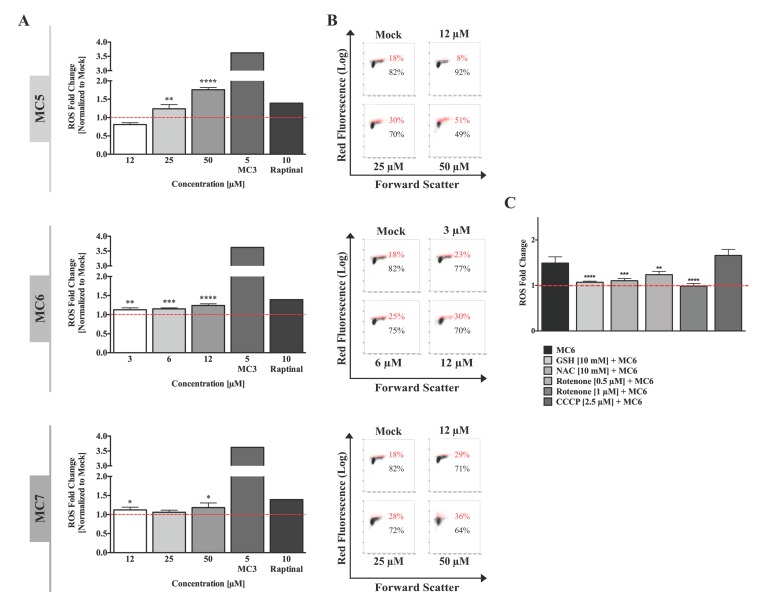
Total cellular reactive oxygen species (ROS) levels are moderately increased by naphthalimide-NHC analogues. (**A**) HCT116 cells were treated with various concentrations of the respective compound for 24 h, after which flow cytometric analysis of ROS generation was performed using the superoxide indicator, dihydroethidium (DHE). A 24 h treatment with the gold(I) NHC complex, MC3 [13] as well as the rapid apoptosis inducer, raptinal [16] was included as positive control. Cellular ROS levels were found to be concentration-dependently induced in response to all the three complexes, with the metal-free ligand (MC5) showing the highest induction. Data were normalized to mock (0.1% DMSO) treatment. Error bars ± SD; *n* = 4. Statistical significance between the respective treatment and mock was determined by two-tailed student’s *t*-test. (**B**) Representative density plots of one out of four biological replicates shown in (**A**); (**C**) ROS levels induced by MC6 (6 μM, 24 h) were found to be significantly decreased in HCT116 cells pre-treated for 1 h with the anti-oxidants, *N*-acetyl-l-cysteine (NAC) and glutathione (GSH), as well as the mitochondrial complex I inhibitor, rotenone at the concentrations indicated. A 2 h co-treatment with the mitochondrial uncoupling reagent (CCCP), and Ru(II) complex (MC6) caused a mild increase in the latter’s effects on ROS generation stained by DHE. Data were normalized to the values of mock (0.1% DMSO) as well as the corresponding single treatments. Statistical significance between MC6 in the absence/presence of each of the inhibitors was determined by two-tailed student’s *t*-test. *, **, ***, and **** represent *p*-values less than or equal to 0.05, 0.01, 0.001, and 0.0001, respectively.

**Figure 4 ijms-19-03964-f004:**
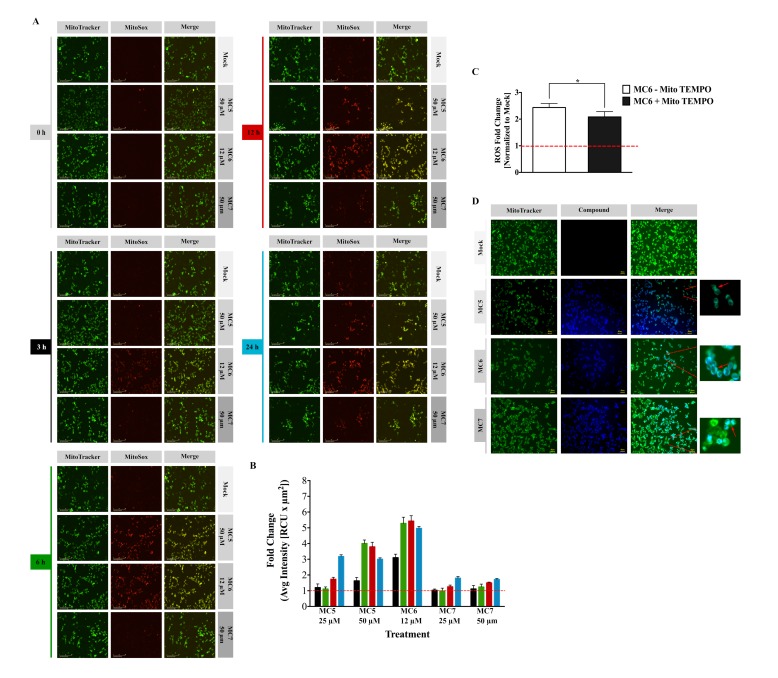
Mitochondrial ROS (mtROS) generation is strongly influenced by naphthalimide-NHC analogues. (**A**) Live cell imaging of mitochondrial superoxide generation stained with MitoSox Red and associated quantification (**B**) as described in the methods’ section. MitoTracker Green was used to indicate mitochondria; (**C**) The mitochondria-targeted anti-oxidant, Mito TEMPO, attenuated the ROS induced by 12 μM of MC6, determined by flow cytometric analysis of MitoSox Red staining. Mito TEMPO (10 μM) was pre-incubated with HCT116 cells 2 h before the exposure to MC6 for 24 h. Data are shown as mean ± SD of three biological replicates. Comparison of ROS fold change between the two groups was performed by two-tailed student’s *t*-test where a *p*-value less than or equal to 0.05 is denoted by *; (**D**) Fluorescence micrographs showing mitochondrial localization of the three complexes in HCT116 cells upon treatment with MC5 (50 μM), MC6 (12 μM), and MC7 (50 μM) for 4 h. Mitochondria were stained by MitoTracker Green. Scale bar: 40 μm. 0.1% DMSO was used as mock.

**Figure 5 ijms-19-03964-f005:**
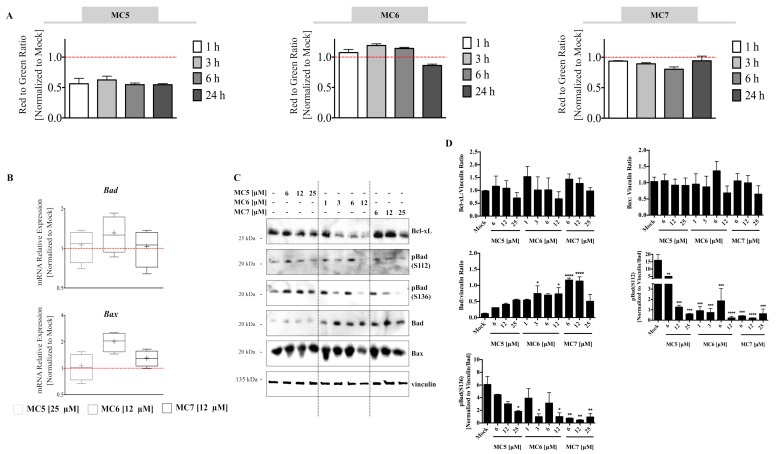
Naphthalimide-NHC analogues reduce, induce and have no clear effect on the Mitochondrial Membrane Potential (MMP) (Δψm) in HCT116 cells. (**A**) MMP was time-dependently decreased by MC5 (25 μM); it was increased in the earlier time points by MC6 (12 μM) followed by a decrease at 24 h; and was found to be mostly unaltered in response to MC7 (25 μM). Error bars are the SD of three biological replicates; (**B**) mRNA expression analysis of the pro-apoptotic Bcl-2 family members, *Bad* and *Bax*, after 24 h of treatment with the compounds at indicated concentrations. Relative expression was calculated by the ∆∆ Ct method where the Ct values of the target genes were normalized to those of vinculin. Lower and upper ends of the bars denote the minimum and maximum values, respectively and “+” in the middle represents the mean. Error bars ± SD; *n* = 4; (**C**) 24 h of treatment with the respective compound led to a concentration-dependent decrease and increase in Bcl-xL and Bad protein levels, respectively, while it had no clear effect on Bax protein expression. Additionally, different phosphorylated forms of Bad were found to decrease in response to treatment; (**D**) Densitometric quantification of Bcl-2 family members normalized to the respective loading control (vinculin). The values of phosphorylated Bad were normalized to those of vinculin as well as total protein levels. Data are expressed as mean ± SEM of two independent experiments, one of which is presented in (**C**). Statistical comparisons were made between mock (0.1% DMSO), and the respective treatment using two-tailed student’s *t*-test. *, **, ***, and **** represent *p*-values less than or equal to 0.05, 0.01, 0.001, and 0.0001, respectively.

**Figure 6 ijms-19-03964-f006:**
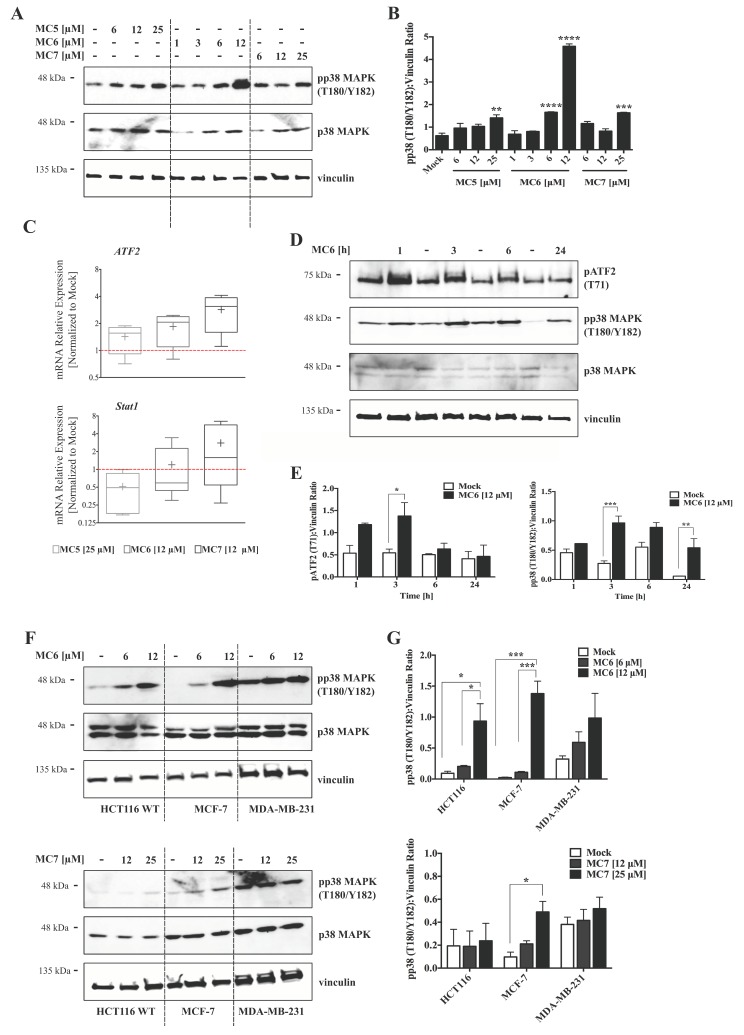
Naphthalimide-NHC derivatives activate the p38 pathway in human breast- and colon cancer cell lines. (**A**) Immunoblots as well as densitometric quantification (**B**) showing the accumulation of pp38 mitogen activated protein kinase (MAPK) (T180/Y182) protein levels by the three complexes in HCT116 CRC cells upon 24 h treatment with increasing concentrations of the respective compound as indicated. The induction appeared to be more profound in case of the Ru(II) analogue, MC6. Data in (**B**) are presented as mean ± SEM of three independent experiments, one of those is shown in (**A**); (**C**) qRT-PCR analysis of p38-associated signaling molecules, *ATF2* and *Stat1* in HCT116 cells treated with the compounds at indicated concentrations for 24 h. Lower and upper ends of the bars denote the minimum and maximum values, respectively, with the “+” sign representing the mean of four biological replicates. Error bars ± SD; (**D**) Time-course analyses of pp38 MAPK (T180/Y182) as well as its down-stream effector, pATF2 (T71) in HCT116 cells treated with 12 μM of MC6, determined by immunoblotting; (**E**) Densitometric analyses of pATF2 (T71) and pp38 (T180/Y182) bands obtained from three independent experiments, one of which is depicted in (**D**). Error bars ± SEM; (**F**) Regulation of p38 MAPK signaling was compared among HCT116, MCF-7 and MDA-MB-231 cancer cell lines treated for 24 h with the metal-containing analogues (MC6 and MC7). In case of MDA-MB-231 where the basal levels of pp38 (T180/Y182) are high, treatments did not profoundly impact the molecule’s phosphorylation; (**G**) Densitometric quantifications illustrate no significant change in pp38 (T180/Y182) levels in MDA-MB-231 cells. Error bars ± SEM; n = 3. Statistical comparisons were made between mock (0.1% DMSO) and the respective treatment using two-tailed student’s *t*-test. *p*-values less than or equal to 0.05, 0.01, 0.001, and 0.0001 are indicated as *, **, ***, and ****, respectively.

**Figure 7 ijms-19-03964-f007:**
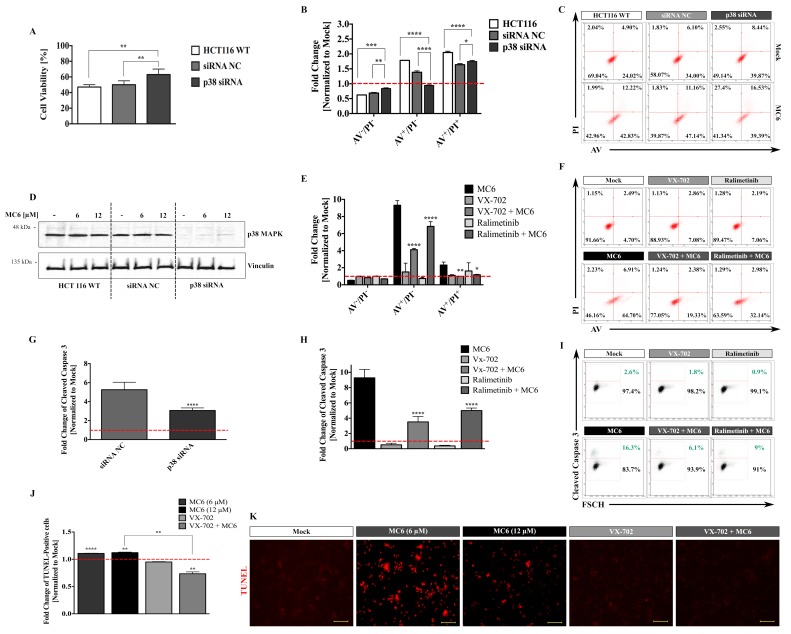
p38 signaling appears to be crucial for the MC6-mediated cytotoxic- and pro-apoptotic effects. (**A**) siRNA-mediated repression of *p38α* was found to hamper the growth inhibitory effects of MC6 in HCT116 cells treated with 12 μM of the compound for 24 h, measured by SRB assay. Percentage cell viability was calculated by normalizing the values of MC6-treated cells to those of the corresponding mock (0.1% DMSO) treatments. Error bars ± SD; *n* = 3; (**B**) Knock-down of *p38α* attenuates the pro-apoptotic response to MC6 (12 μM, 24 h), assessed by flow cytometric analysis of AV/PI staining. Percentage cell population in each quadrant was normalized to the respective mock (0.1% DMSO) treatment. Error bars ± SD. Multiple comparisons were performed using two-way ANOVA followed by a post-hoc Tukey test; (**C**) Density plots representative of three biological replicates illustrate increased population of AV^+^/PI^−^ and AV^+^/PI^+^ with treatment, however, to a lesser extent in case of cells transfected with anti-*p38α* siRNA; (**D**) Confirmation of knock-down efficiency, as determined by immunoblotting; (**E**) chemical inhibition of p38α abrogates the MC6-mediated apoptosis. HCT116 cells were treated with 12 μM of MC6 in the absence/presence of p38α inhibitors, VX-702 and Ralimetinib at a concentration of 0.5 μM for 24 h. Error bars ± SD, *n* = 3. Asterisks show significance in the amount of early- and late apoptotic population between cells treated with MC6 and each of the two inhibitors, and MC6 as a single agent, determined by two-tailed student’s *t*-test; (**F**) Representative density plots of one out of three biological replicates demonstrate reduced number of AV^+^/PI^−^ and AV^+^/PI^+^ cells when p38α activity is inhibited; (**G**) Flow cytometric analysis of caspase 3 activation shows significantly less cleaved caspase 3 expression in cells transfected with anti-*p38α* siRNA as compared to that of the negative control. Error bars ± SD, *n* = 6. Statistical significance between the two groups was made using two-tailed student’s *t*-test; (**H**) Chemical inhibition of p38α was found to decrease the levels of active caspase 3 in a similar manner to that of *p38α* knock-down. Error bars ± SD, *n* = 6. Statistical comparison was performed between combination treatments and MC6 using two-tailed student’s *t*-test; (**I**) Representative density plots of one out of six biological replicates; (**J**) Detection of apoptotic cells using TUNEL assay. HCT116 cells were treated with the indicated concentrations of MC6 for 24 h in the absence/presence of VX-702 (0.5 μM). Statistical significance was calculated between mock and the respective treatment as well as MC6 as single agent and in combination with VX-702 using two-tailed student’s *t*-test; (**K**) Representative fluorescence images of TUNEL reaction. Scale bar: 100 μm. *p*-values less than or equal to 0.05, 0.01, 0.001, and 0.0001 are denoted as *, **, ***, and ****, respectively.

**Figure 8 ijms-19-03964-f008:**
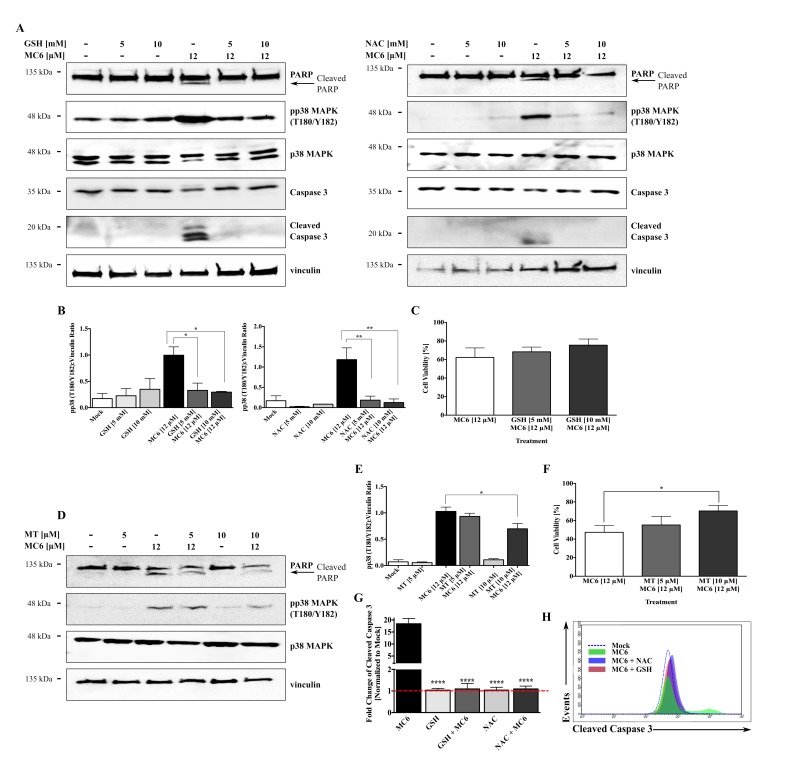
MC6-induced cytotoxicity and pro-apoptotic effects are mediated via the ROS-p38 signaling axis. (**A**) Treatment with GSH and NAC at the indicated concentrations 1 h prior to the addition of MC6 (12 μM) for 24 h blocked the activation of p38 as well as cleavages of caspase 3 and PARP, detected by immunoblotting; (**B**) Densitometric analyses show a significant reduction in the MC6-induced pp38 (T180/Y182) accumulation in the presence of anti-oxidants. Error bars indicate the SEM of three independent experiments, one of those is presented in (**A**); (**C**) Increased cellular survival of HCT116 cells pre-treated with either 5 or 10 mM of GSH 1 h before the addition of MC6 (12 μM) for 24 h, as determined by SRB assay. Data represent mean ± SD of three biological replicates, normalized to mock (0.1% DMSO) and the respective GSH treatment; (**D**–**F**) HCT116 cells pre-incubated with either 5 or 10 μM of Mito TEMPO (MT) for 2 h were treated with MC6 (12 μM) for 24 h. The mitochondria-targeted ROS scavenger was found to attenuate the MC6-mediated p38 activation as well as PARP cleavage at the highest used concentration (10 μM), as detected by immunoblotting (**D**) and the associated densitometric quantification (**E**), obtained from three independent experiments. Error bars ± SEM. Additionally, it rescued the MC6-mediated cytotoxic effects, as determined by SRB assay (**F**). Percentage cell viability of MC6-treated cells was normalized to mock (0.1% DMSO) and the respective MT treatment. Error bars ± SD; *n* = 3; (**G**) Flow cytometric analysis of caspase 3 activation illustrates significantly lower levels of the cleaved form of the protein in cells pre-treated for 1 h with either GSH (10 mM) or NAC (10 mM) as compared to that of MC6-treated cells (12 μM, 24 h). Error bars ± SD, *n* = 3; (**H**) Representative histogram of one out of three biological replicates presented in (**G**) demonstrates a left-ward shift in caspase 3 activity in the presence of ROS scavengers. Statistical significance between the MC6-treated cells in the absence/presence of anti-oxidants was calculated using two-tailed student’s *t*-test. *, **, ***, and **** on the figures represent *p*-values that are less than or equal to 0.05, 0.01, 0.001, and 0.0001, respectively.

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
