# Peer review of "A Ruthenium(II) N-Heterocyclic Carbene (NHC) Complex with Naphthalimide Ligand Triggers Apoptosis in Colorectal Cancer Cells via Activating the ROS-p38 MAPK Pathway"

_ijms, 2018, doi:10.3390/ijms19123964_

Reviewer 1 Report

“A Ruthenium (II) N-heterocyclic Carbene (NHC) Complex with Naphthalimide Ligand Triggers Apoptosis in Colorectal Cancer Cells via Activating the ROS-p38 MAPK Pathway” is an interesting study focusing on the anti-cancer properties of naphthalimide-NHC compounds. The authors demonstrated that mitochondrial derived reactive oxygen species (ROS) plays a critical role in the cytotoxicity of naphthalimide-NHC compounds in cancer cells and is through the activation of p38-MAPK signaling pathway. The manuscript is well-written, and the findings can be relevant and interesting for researchers in the field of cancer biology. The authors are recommended to check for grammatical errors and look at the flow of manuscript.

Author responds: We thank the reviewer for the positive feedback. As suggested, we checked for grammatical and linguistic errors along the manuscript and highlighted all the changes for your convenience

Reviewer 2 Report

In the manuscript ijms355150 entitled “A Ruthenium(II) N-heterocyclic Carbene (NHC) Complex with Naphthalimide Ligand Triggers Apoptosis in Colorectal Cancer Cells via Activating the ROS-p38 MAPK Pathway", the effect of this NHC compounds on proliferation and survival of colorectal cancer cells have been evaluated. The authors propose that they have a pro-apoptotic effect acting through a mechanism dependent on ROS and p38a MAPK. However, their conclusions are only partially supported by the results. Therefore, additional experiments will be necessary. Moreover, other analyses and changes along the manuscript are required to improve it according to the following comments:

1-In figure 1a, it is not explained whether it is only used a single control of untreated cells at time 0 or different controls at each time point. It would be necessary to have a control for each time and then to refer all the data (from untreated and treated cells) in relation to the control value at time 0. In this way, it would be possible to understand why, in some cases (i.e. MC5 6mM in HCT116 and MDA-MB231), the viability after 24h of treatment with the MC compounds is higher than 100%. This is difficult to understand unless cells are proliferating. Therefore, it is important to show all the controls, so that it could be seen if cells treated with MC compounds have a higher or a lower viability than the corresponding control at each time.

Author responds: The %cell viability was calculated by dividing the absorbance of the treated cells over the absorbance of the respective mock treatment (with one mock treatment for each time point). However, as the reviewer suggested the “%cell viability” was changed to “absorbance [535#nm]”, so the control values for each of the time points can be depicted. As shown in Figure'1b, the absorbance of mock treatment increases over time, therefore the cellular viabilities higher than 100% appear to be due to the proliferation of cells with time. Accordingly, Figure'S1 was changed in a similar manner to Figure'1 where %cell viability was replaced by “absorbance [535 nm]”.

2-On page 3 line 114, the authors indicate that they examine the transcriptional activity of p21, when they are just quantifying the levels of p21 mRNA. This should be corrected. In addition, it would be important to determine p21 protein levels to be sure if p21 could be mediating the cell cycle arrest. In addition, regarding figure 2a, it would be necessary to increase the size of the images.

Author responds: The “transcriptional activity of p21” was corrected as the “mRNA levels of p21”; this change is highlighted in the text (line 115). Furthermore, p21 protein levels were determined in HCT116 cells treated with the compounds for 24 h using immunoblotting/densitometric quantification. As shown in Figure'2d and Figure'2e, p21 protein expression was found to be induced in response to all the MC compounds. As suggested by the reviewer, the size of the histograms in'Figure'2a'was increased for better readability.

3-The results shown in figure 3 are not clear. Although the authors indicate that there is a consistent increase in ROS, this is only observed for MC5 compound at 50mM. In any case, it should be necessary to show the statistic.

Author responds: The statistical significance between mock (0.1% DMSO) and the respective treatments was calculated using two-tailed student’s t-test and is denoted as asterisks in Figure'3a.

4-The images shown in figure 4A and 4D are very small, so it is not so easy to see the changes described in the text. In addition, the decrease in ROS upon Mito TEMPO treatment in cells treated with 12mM MC6 is quite modest to justify that ROS generated in the mitochondria are so important as the authors say.

Author responds: The sizes of the fluorescence micrographs in Figure'4a and 4d were increased for better readability. Although significant, the ability of Mito TEMPO to decrease mtROS induced by MC6 appears to be modest as the reviewer mentioned. There are a variety of methods in order to identify ROS generated from mitochondria among others are the fluorescence probes attached to lipophilic cations which are attracted to the negative potential environment generated by the electrochemical gradient across the inner mitochondrial membrane; for example, MitoSOX for probing mitochondrial superoxide species [Mukhopadhyay P. et al. Nat Protoc (2007), 2:2295–2301], and MitoPY1 for the detection of mitochondria-derived H2O2 [Dickinson B.C. et al. Am Chem Soc 2008, 130:9638–9639]. In this study, we observed a clear difference between the total cellular ROS levels and mtROS, as determined by DHE and MitoSOX Red fluorescence, respectively. For instance, as shown in Figure'3a, FACS analysis of DHE staining shows only a ~1.2Xfold induction after 24 h of MC6 treatment whereas upon MitoSOX Red staining the fold change of ROS generation reaches to ~5Xfold (Figure'4b). We therefore hypothesized that the MC6Xinduced ROS may be generated from mitochondria, and the decrease in mtROS levels in cells preXtreated with Mito TEMPO further proves the source of ROS.

In addition to this, several reports have suggested the combination of conventional ROS detection methods (e.g. DHE staining) together with electron transport chain (ETC) inhibitors in order to identify mtROS. In the revised manuscript, we tested the complex I inhibitor, rotenone, and found that co-treatment with rotenone and MC6 attenuates the latter’s effects on total cellular ROS production (Figure'3c). Taking all the above-mentioned data into consideration, it is most likely that the ROS induced by MC6 is produced from mitochondria. We agree that further investigations are of interest to better understand the mechanism by which ROS are generated. Although other mitochondria-specific ROS scavengers (e.g. MitoQ) and ETC inhibitors (i.e. complex II and III inhibitors) could be used, we feel that the use of Mito TEMPO and the profound decrease of MC6-induced ROS with rotenone clearly demonstrates the contribution of mitochondria in generation of ROS. Nevertheless, more detailed investigation by means of further discriminating inhibitors as well as assessment of mitochondrial anti-oxidant systems (e.g. SOD2) could be the task for a future study.

5-Figure 5A is extremely small to see anything, so that it should be enlarged. In addition, the histograms from figure 5B showing the changes in MMP are not too informative. Therefore, it is not clear why the authors show these results in this figure. It should be better to show them as a supplementary figure. On the other hand, and more important, it is unclear why they are only measured the levels of Bad and Bax mRNAs, but not those of the proteins. It is more important to measure Bax and Bad protein levels and P-Bad levels to have a good idea of its potential involvement in an apoptotic process. 

Author responds: As the reviewer suggested, the histograms were enlarged and moved to the supplementary material (Figure'S4). Additionally, the size of the MMP graphs was increased (Figure'5a). In addition to the mRNA expression, protein levels of the pro-apoptotic BclX2 members, Bad and Bax, were determined using immunoblotting. As depicted in Figure'5c and 5d, Bax protein expression was found to be mostly unaltered in response to the MC compounds. Bad protein levels, on the other hand, were significantly up-regulated in particular upon treatment with the metal-containing analogues, MC6 and MC7. Importantly, we evaluated three different phosphor-isoforms of Bad; pBad (S112), pBad (S136), and pBad(S155) and found decreased phosphorylation of the first two with treatment (Figure'5c and 5d). The latter phosphor-Bad was not found to be detectable in HCT116 cells in the absence/presence of naphthalimide-NHC analogues. Phosphorylation of Bad at the aforementioned sites has been reported to disrupt the interaction of the protein with the pro-survival factors, BclX2 and BclXxL, which consequently inhibits its death-promoting activity. These findings imply the involvement of Bad in promoting cell death in response to the compounds.

6-On page 8, lines 229, 240 and in other places along the text, the authors say “protein expression of phospho-p38MAPK”, which is incorrect. It should be described as an increase in the levels of phospho-p38MAPK. In fact, it is likely that p38 is phosphorylated in response to these compounds, but its expression is not necessarily higher. In any case, total levels of p38 MAPK should be determined for figure 6A, 6D and 6F and for other figures.

Author responds: We like to thank the reviewer for this critical remark#and#for# the additional value obtained by including the total p38 protein levels in addition to the analysis of p38 phosphorylation. As the reviewer pointed out, we have changed the “protein expression of phosphor-p38 MAPK” to “levels of phosphor-p38 MAPK”, which has been highlighted in the text. More importantly, total protein levels of p38 MAPK have been determined in Figure'6a, 6d, 6f as well as Figure'7e and 7h. As illustrated in the mentioned figures, the naphthalimide-NHC conjugates were not found to affect total levels of p38 MAPK. The two bands appearing on some of the blots (e.g. Figure'6d and 6f) demonstrate two different isoforms of the protein; p38α (lower band ~40 kDa), and p38β (upper band ~43 kDa), since the antibody is not specific to individual isoforms (Santa Cruz Biotechnology, #SC7972). It has to be noted that p38α is the most abundant p38 MAPK in most cell types including HCT116.

7-It is unclear the meaning of Stat1 mRNA levels (fig 6C). The protein levels of phospho-Tyr Stat1 and total Stat1 should be determined.

Author responds: The mRNA levels of the genes, ATF-2 and Stat1, was determined as two examples of several p38 downstream targets (TP53, TAB1, MNK1/2 to name a few more) in order to further indicate the activation of the signaling pathway. Although the status of Stat proteins in response to the MC compounds is not the focus of this study, we have tried to look into the tyrosine (Y701)- as well as serine (S727) phosphorylation of the protein as the reviewer suggested. The latter phosphorylation site was thought to be particularly interesting since it is known to be phosphorylated by p38 MAPK. However, we were not able to detect phosphorylated Stat1 in HCT116 cells upon treatment with the naphthalimide-NHC conjugates. Additionally, we tested the protein levels of total and phosphor-Stat3 (tyrosine/serine residues; Y705/S727). While phosphor-Stat3 protein levels were significantly higher than that of phosphor-Stat1, no clear regulation was observed in response to the three analogues (data not shown).

8-The results derived from annexin V/PI analysis shown in Figure 7C doesn’t indicate that early apoptosis induced by MC6 is attenuated upon p38a silencing, as authors described. The apoptosis (% of cells AV+/PI-) found in the control upon treatment with MC6 is 42.8%, in the control performed with a siRNA NC 47.14% and in cells with siRNA p38 39.39%, which is the same value than that found in untreated p38 silenced cells (39,87%) and very similar to the one found in siRNA NC cells (34,66%). According to all this, basal levels of apoptosis are increased in cells transfected with siRNA NC and siRNA p38 and then, there is a slight increase in apoptosis in response to MC6. This indicates that the transfection with either siRNA NC or siRNA p38 is inducing a pro-apoptotic effect. Therefore, in order to conclude that p38a mediates the pro-apoptotic effect of MC6, other experiments should be done using a p38 inhibitor and/or a permanent p38a Knock-down. Moreover, apoptosis should be determined using other complementary approaches such as the quantification of active caspase 3 (cleaved form).

Author responds: In order to complement the results obtained from p38 knock-down experiments, we included two well-known chemical inhibitors of p38α, VXX702 and Ralimetinib (LY2228820), and measured apoptosis using FACS analyses of both AV/PI staining as well as cleaved caspase 3 quantification. As shown in Figure'7e, 7f, 7h, and 7i, upon chemical inhibition of p38α the amount of apoptotic cells is significantly reduced, as determined by AV+ cells and cleaved caspase 3 levels. Additionally, siRNA-mediated repression of p38α (MAPK14) resulted in a significant decrease in the cleaved caspase 3 expression (Figure'7g), altogether suggesting less apoptotic response to MC6 in the absence of p38α signaling.

9-In order to be sure about the effect of the antioxidants (Figure 7), it should be determined the levels of active caspase 3 as the blots showing PARP cleavage are not clear.

Author responds: As the reviewer mentioned, we added total/cleaved caspase 3 protein levels to the blots in Figure'8a. Additionally, caspase 3 activation was determined in the absence/presence of NAC and GSH using flow cytometry (Figure'8g and 8h). The results showed a consistent reduction in cleaved caspase 3 levels in the presence of ROS inhibitors, further confirming the involvement of ROS in the MC6-mediated pro-apoptotic effects.

Minor points:

1-On page 3 line 107, after measuring cell viability in Fig. 1a, the authors define the effect of the MC compounds as an inhibitory activity on proliferation. Obviously, this is not correct taking into account the performed experiment. After doing the cell cycle analysis it would be correct, but not before. Therefore, the text should be accordingly modified.

Author responds: As the reviewer pointed out, this term was corrected to “cytotoxicity” or “cytotoxic effects” along the text and is highlighted for your convenience.

2-Axes from figure 7C should be labeled.

Author responds: The labels (AV/PI) were added to Figure'7c.

3-Additional references should be included when referred to the role of p38 in cisplatin action on CRC cells such as that from Bragado et al. (2007) Apoptosis, 12(9):1733-42.

Author responds: We have discussed the mentioned study and added two other references [Abdellah Mansouri et al. (2003) J Biol Chem, 278(21):19245-56, and C. St Germain et al. Neoplasia, 12(7) (2010) 527-38], showing the role of p38 activation in the anti-tumor response of cisplatin, which is also highlighted in the text.

Reviewer 3 Report

The authors provided evidence that NHC complex with naphthalimide ligand triggers apoptosis in cancer cells through ROS-activated p38-MAPK signaling pathway. However, different effects were observed among three kinds of NHC complex with naphthalimide ligands (MC5, MC6, and MC7). These results cannot conclude the cell death caused by these compounds to be predominately through ROS-activated p38-MAPK signaling pathway. More important, the apoptotic mechanism through ROS-activated p38-MAPK pathway is well documented in cancer cells. The reviewers cannot support the findings of the present study is novel and interesting in cancer therapy.

Author responds: As the reviewers explained, we observed different effects regarding ROS induction as well as p38α activation among the three naphthalimide-NHC analogues. However, the study has been focused on the Ru(II) complex (MC6) due to its highest induction  of mtROS as well as p38α activation, and thus the involvement of ROS-p38 signaling axis in the cytotoxic and pro-apoptotic effects has been only investigated and is only concluded for this derivative as clearly mentioned in the title/abstract and elsewhere in the text and conclusion. Several lines of studies have documented the pro-apoptotic role of ROS-activated p38 MAPK in different types of tumor models as the reviewers pointed out. Despite this, to our knowledge the importance of the aforementioned pathway has not been previously investigated as a target of naphthalimide-NHC conjugates. Additionally, p38 signaling appears to have a dual role in cancer therapy since its inhibition and activation are both beneficial depending on the onset of cellular transformation, cancer type as well as the status of other MAPKs. Therefore, evaluating the pro- or anti-tumorigenic potential of p38α in different cellular contexts and in response to novel small molecules such as the compounds of the present study may add to our understanding of its role in cancer treatment. In the revised manuscript, we tested the effects of two well-established chemical inhibitors of p38α, VXX702 and Ralimetinib, in combination with MC6 (Figure'7e&i). Furthermore, we have tested the effects of ROS scavengers (NAC and GSH) and mitochondrial respiratory chain inhibitors (rotenone and CCCP) on the MC6-mediated total cellular ROS generation (Figure' 3c), and tried alternative methods to determine apoptosis in response to MC6 in the absence/presence of anti-oxidants (Figure'8a, 8d, 8g, and 8h) to further support our conclusion with regards to the involvement of ROS-p38α axis in MC6-mediated apoptosis. In addition to this, the activity of the other two MAPK family members, ERK and JNK, was assessed using phosphor-protein ELISA-microarray analysis as well as immunoblotting, which showed only a mild reduction in case of the former and no significant regulation in case of the latter (Figure'S5).

Minor:

The full name should be given in the first place of the abbreviation. The standard compound used in the experiments should be described in the ligand of the figures.

Author responds: As mentioned, we have checked all the abbreviations and added the full names in the first place they appeared in the text as well as in the abbreviation section. Additionally, we described and cited the standard compounds we used as reference in the figure legends, which is also highlighted for your convenience.

Reviewer 4 Report

Authors suggest that naphthalimide-NHC conjugates with the Ru(II) analogue (MC6 attenuated the anti-proliferative and pro-apoptotic effects of MC6 in HCT116 cells via ROS mediated p38 activation. Despite interesting data, it has some concerns as follows:

How about cytotoxicity of MC6 in normal cells

Author responds: We have tested the cytotoxicity of the three analogues against human foreskin fibroblasts (HFFs) using SRB assay. As shown in Figure'S1, HFF cells were not found to be sensitive to the MC compounds 6 at the indicated concentrations/time points, suggesting preferential toxicity of these compounds towards cancer cells.

How about solubility of this compound?

Author responds: All the physicochemical properties of the presented compounds have been previously reported: Streciwilk W et al. Eur J Med Chem, 156:148-161 and Streciwilk W et al. ChemMedChem, 12(3):214-225. It has to be noted that in the abovementioned articles, MC5, MC6, and MC7 are designated as 4c, 6c, and 5c, respectively. Importantly, all the compounds were sufficiently soluble under the described assay conditions.

Show effect of MC6 on TUNEL positive cells.  Add total form of p38.

Author responds: To further evaluate the pro-apoptotic response of HCT116 cells to MC6, we used alternative methods for the detection of apoptosis including, quantification of caspase 3 activation (Figure'7e&i) as well as TUNEL analysis (Figure'7k and 7j) in the absence/presence of p38! activity, as the reviewer suggested. Total p38 MAPK levels were added to the following immunoblots; Figure'6a, 6d, 6f as well as Figure'7e and 7h.'As illustrated in the mentioned figures, the naphthalimide-NHC conjugates were not found to affect the total protein levels of p38 MAPK. The two bands appearing on some of the blots (e.g. Figure'6d'and 6f) demonstrate two different isoforms of the protein; p38α (lower band ~40 kDa), and p38β (upper band ~43 kDa), since the antibody is not specific to individual isoforms (Santa Cruz Biotechnology, SC7972). It has to be noted that p38α is the most abundant p38 MAPK in most cell types including HCT116.

How about effect on ERK and JNK

Author responds: The considerable crosstalk between p38 and other MAPKs (ERK and JNK kinases) is known to influence the overall outcome of the pathway activation. We therefore evaluated the phosphorylation status of these two kinases upon treatment with the MC compounds using phosphor-protein ELISA- microarray analysis as well as immunoblotting, as the reviewer suggested. As shown in Figure'S5, a modest reduction was observed in the activated form of Erk1/2, pErk1/2 (T202/Y204), which however failed to reach statistical significance. This reduction may be due to the reduced total protein levels, as determined by immunoblotting (Figure'S5a), and the associated densitometric quantification (Figure'S5b). Additionally, we tested the levels of phosphor-JNK (T183/Y185) in response to the naphthalimide-NHC analogues and found extremely low expression of the protein in HCT116 cells with no significant regulation upon treatment (Figure'S5c). Altogether, it is most likely that the mode of in vitro anti-cancer activity of the presented compounds occurs independent of the other two members of MAPK family.

Show effect of p38 or ROS inhibitor on MC6 induced apoptosis

Author responds: As shown in Figure' 7, the MC6-induced apoptosis has been investigated with and without genetic/pharmacological inhibition of p38α using different methods including AV/PI staining (Figure 7a&f), flow cytometric analysis of caspase 3 activation (Figure'7g&i) as well as TUNEL assay (Figure'7j and 7k). Based on our results, either chemical- or genetic inhibition of p38α using two well-known inhibitors (VXX702 and Ralimetinib) and RNA interference, respectively led to a significantly smaller number of apoptotic cells in response to MC6, highlighting the important role of p38α in the molecule’s apoptogenic activity. In addition to p38α, we have evaluated the influence of ROS scavengers (GSH, NAC, and Mito TEMPO) on the pro-apoptotic effects of MC6 using western blot analyses of PARP and caspase 3 cleavages (Figure'8a&d) as well as FACS analyses of cleaved caspase 3 expression (Figure'8g&h). The results showed that either of the mentioned anti-oxidants are able to protect HCT116 cells against MC6-mediated apoptosis, further confirming the involvement of ROS in the observed anti-cancer effects of the molecule.

How about its effect in p53 mutant type cancer cells, since HCT116 and MCF7 cells are p53 wild type cancer cells.

Author responds: p38 MAPK is known to enhance the pro-apoptotic response of p53 by phosphorylating the protein at two serine residues (Ser 33/46). p53, on the other hand, is a redox-sensitive protein whose activity is majorly influenced by intracellular ROS levels. In addition to this, p53 may act upstream of ROS induction with inducing both pro- and anti-oxidant genes depending on the severity and type of the stress. Therefore, it is most likely that the proposed mechanism for MC6 (via the ROS-induced activation of p38α signaling) varies in different p53 statuses, as the reviewer pointed out. Although the effects of different p53 variations is not the focus of the present study, we have briefly evaluated the cytotoxic efficacy of the naphthalimide-NHC analogues in cells possessing deficient- and mutant p53 by including p53-null HCT116 as well as HT-29 CRC cells, respectively. As shown in Figure'S2, HT-29 appeared to be the most resistant cell line at most tested conditions. With regards to p53-negative HCT116 cells, there was no consistent difference compared with its wild-type (WT) counterpart, despite reaching statistical significance at some of the concentrations/time points. Further investigations are necessary to have a clear picture regarding the chemotherapeutic efficacy of the three analogues in different p53 statuses.

Round  2

Reviewer 2 Report

The revised version of the manuscript entitled “A Ruthenium(II) N-heterocyclic Carbene (NHC) Complex with Naphthalimide Ligand Triggers Apoptosis in Colorectal Cancer Cells via Activating the ROS-p38 MAPK Pathway", includes new data and comments supporting the main conclusions. Thus, it is now demonstrated that the NHC compounds decrease proliferation and survival of colorectal cancer cells, while increasing apoptosis. This pro-apoptotic effect is dependent on a ROS-p38a MAPK pathway.

 Author Response

Point 1: The revised version of the manuscript entitled “A Ruthenium(II) N-heterocyclic Carbene (NHC) Complex with Naphthalimide Ligand Triggers Apoptosis in Colorectal Cancer Cells via Activating the ROS-p38 MAPK Pathway", includes new data and comments supporting the main conclusions. Thus, it is now demonstrated that the NHC compounds decrease proliferation and survival of colorectal cancer cells, while increasing apoptosis. This pro-apoptotic effect is dependent on a ROS-p38a MAPK pathway.

Response 1: We would like to thank the reviewer for the positive feedback, and the previous comments and suggestions which helped us to improve the manuscript and provide further data in order to support the main findings of this study.

Reviewer 4 Report

Much improved, but we cannot clearly understand important role of ROS mediated pp38 MAPK  in HCT166 cells.

How about effect of MC6 on pJNK and pERK in HCT166 cells.

How about effect of NAC and p38 inhibitor on PARP cleavage and caspase3 in HCT166 cells.

Author Response

Before addressing the following points, we would like to thank the reviewer for raising important issues, which helped us to include and discuss the critical points in the revised version of the manuscript.  

Point 1: How about effect of MC6 on pJNK and pERK in HCT166 cells.

Response 1: For the effects of MC6 (and the other two naphthalimide-NHC derivatives) on the activity of phospho-ERK and phospho-JNK, please see Figure S5 (page 22) and page 8 lines 278-286, highlighted in yellow. Additionally, in the revised manuscript we have included this issue both in the introduction (page 2, lines 73 and 74) and discussion (page 13, 431-438) to further emphasize on the importance of the crosstalk between different members of the MAPK family, as mentioned by the reviewer. These changes are highlighted in grey in the text.

As shown in Figure S5a-c, the phosphorylation status of these two kinases has been evaluated using immunoblotting and phospho-protein ELISA-microarray analysis. Based on our results, there is a modest reduction in the activated form of ERK1/2 (pERK1/2 (T202/Y204)), which failed to reach statistical significance. This reduction may be due to the reduced total protein levels, as determined by immunoblotting (Figure S5a) and the associated densitometric quantification (Figure S5b). Additionally, we tested the levels of phospho-JNK (T183/Y185) in response to the MC compounds and found extremely low protein expression in HCT116 cells with no clear regulation upon treatment (Figure S5c) in comparison to that of the positive control (cisplatin). Altogether, it is most likely that the mode of in vitro anti-cancer activity of the presented compounds occurs independent of the other two members of MAPK family.

Point 2: How about effect of NAC and p38 inhibitor on PARP cleavage and caspase 3 in HCT166 cells.

Response 2: Please see Figure 8a (page 12) and page 11 lines 364-366 (highlighted in grey) for the inhibitory effect of ROS scavengers (NAC and GSH) on cleavages of PARP and caspase 3 induced by MC6, determined by immunoblotting. Additionally, cleaved caspase 3 levels in the presence/absence of the aforementioned ROS inhibitors were quantified using flow cytometry (please see Figure 8g and 8h, and page 11 lines 371-373). The results clearly demonstrate that either of these anti-oxidants is able to protect HCT116 cells against MC6-mediated PARP/Caspase 3 cleavage; two hallmarks of apoptotic cell death.

The effects of genetic and chemical inhibition of p38a on MC6-mediated cytotoxic- and pro-apoptotic effects are demonstrated in Figure 7 (page 10), evaluated by a number of assays including AV/PI staining, flow cytometric analysis of cleaved caspase 3 levels, and TUNEL assay. For the influence of p38 chemical inhibitors on caspase 3 activation please see Figure 7h and 7i (page 10) as well as page 10 lines 324-328, all highlighted in the text.

Round 3

Reviewer 1 Report

They addressed well to my comments.